# Trends in Prevalence of Diabetes among Twin Pregnancies and Perinatal Outcomes in Catalonia between 2006 and 2015: The DIAGESTCAT Study

**DOI:** 10.3390/jcm10091937

**Published:** 2021-04-30

**Authors:** Lucia Gortazar, Juana Antonia Flores-Le Roux, David Benaiges, Eugènia Sarsanedas, Humberto Navarro, Antonio Payà, Laura Mañé, Juan Pedro-Botet, Albert Goday

**Affiliations:** 1Department of Endocrinology and Nutrition, Hospital del Mar, Passeig Marítim, 25-29, 08003 Barcelona, Spain; lucia.gortazar@gmail.com (L.G.); 94066@parcdesalutmar.cat (J.A.F.-L.R.); humberto20192@gmail.com (H.N.); laurams47112@gmail.com (L.M.); 86620@parcdesalutmar.cat (J.P.-B.); Agoday@parcdesalutmar.cat (A.G.); 2Department of Medicine, Universitat Autònoma de Barcelona, 08139 Barcelona, Spain; 3Institut Hospital del Mar d’Investigacions Mèdiques, 08003 Barcelona, Spain; 4Consorci Sanitari de l’Alt Penedès Garraf, 08720 Vilafranca del Penedès, Spain; 5Health Information Management Department, Hospital del Mar, 08003 Barcelona, Spain; esarsanedas@parcdesalutmar.cat; 6Department of Gynecology and Obstetrics, Hospital del Mar, 08003 Barcelona, Spain; apaya@parcdesalutmar.cat

**Keywords:** epidemiology, twins, gestational diabetes, pre-existing diabetes, prevalence, trends, perinatal outcomes

## Abstract

The aims of our study were to evaluate the trends in the prevalence of diabetes among twin pregnancies in Catalonia, Spain between 2006 and 2015, to assess the influence of diabetes on perinatal outcomes of twin gestations and to ascertain the interaction between twin pregnancies and glycaemic status. A population-based study was conducted using the Spanish Minimum Basic Data Set. Cases of gestational diabetes mellitus (GDM) and pre-existing diabetes were identified using ICD-9-CM codes. Data from 743,762 singleton and 15,956 twin deliveries between 2006 and 2015 in Catalonia was analysed. Among twin pregnancies, 1088 (6.82%) were diagnosed with GDM and 83 (0.52%) had pre-existing diabetes. The prevalence of GDM among twin pregnancies increased from 6.01% in 2006 to 8.48% in 2015 (*p* < 0.001) and the prevalence of pre-existing diabetes remained stable (from 0.46% to 0.27%, *p* = 0.416). The risk of pre-eclampsia was higher in pre-existing diabetes (15.66%, *p* = 0.015) and GDM (11.39%, *p* < 0.001) than in normoglycaemic twin pregnancies (7.55%). Pre-existing diabetes increased the risk of prematurity (69.62% vs. 51.84%, *p* = 0.002) and large-for-gestational-age (LGA) infants (20.9% vs. 11.6%, *p* = 0.001) in twin gestations. An attenuating effect on several adverse perinatal outcomes was found between twin pregnancies and the presence of GDM and pre-existing diabetes. As a result, unlike in singleton pregnancies, diabetes did not increase the risk of all perinatal outcomes in twins and the effect of pre-existing diabetes on pre-eclampsia and LGA appeared to be attenuated. In conclusion, prevalence of GDM among twin pregnancies increased over the study period. Diabetes was associated with a higher risk of pre-eclampsia, prematurity and LGA in twin gestations. However, the impact of both, pre-existing diabetes and GDM, on twin pregnancy outcomes was attenuated when compared with its impact on singleton gestations.

## 1. Introduction

Diabetes mellitus (DM) is the most frequent metabolic complication of pregnancy. Pre-existing diabetes and gestational diabetes mellitus (GDM) affect around 0.3–0.6% and 2–10% of pregnancies, respectively, in Europe [1,2,3,4,5], and there is evidence that the prevalence of diabetes in pregnancy is rising worldwide [6,7,8]. Women with GDM and pre-existing diabetes are at increased risk of adverse maternal and neonatal outcomes including congenital malformations, caesarean section, pre-eclampsia, macrosomia and neonatal hypoglycaemia [9,10]. The risk of perinatal complications appears to be greater in pre-existing diabetes than in GDM [11].

Twin pregnancies are also at high risk of maternal and neonatal morbidity and their incidence has increased worldwide in recent decades in association with increasing maternal age and the use of assisted reproductive technology (ART) [12].

It is therefore not surprising that both conditions are more likely to coincide and diabetes might add additional risk to twin gestations. However, evidence of the effect of diabetes on maternal and perinatal outcomes in twin pregnancies is conflicting. In this respect, population-based studies evaluating epidemiological data in twin pregnancies complicated by diabetes in Southern Europe are scant and few have focused on pre-existing diabetes.

Moreover, it has suggested that an interaction between twin pregnancies and glycaemic status might be present. Hiersch et al. reported that unlike in singleton pregnancies, GDM was not associated with hypertensive complications, neonatal intensive care unit admission and neonatal hypoglycaemia in twins [13]. Diabetes might differently influence outcomes in twin and singleton pregnancy, therefore affecting clinical care in women with diabetes and twin gestations.

Thus, our study was aimed to assess trends in prevalence of GDM and pre-existing diabetes among twin pregnancies between 2006 and 2015 in Catalonia, Spain; to evaluate the influence of both diabetic conditions on adverse gestational and perinatal outcomes in twin gestations and to ascertain the interaction between twin pregnancies and glycaemic status.

## 2. Materials and Methods

A retrospective cohort study was conducted using the Minimum Basic Data Set for Hospital Discharge (CMBD-AH). All maternal hospital discharge records of singleton and multiple deliveries in Catalonia from January 2006 to December 2015 were collected. The CMBD-AH database covers more than 95% of private and public hospital deliveries in Spain and is managed by the Spanish Ministry of Health which conducts periodic audits to ensure coding reliability.

Catalonia is an autonomous community in the northeast of Spain. It is the second largest in terms of population (7.7 million inhabitants) and Barcelona, its capital, is the second most populated city in Spain (1.6 million inhabitants) [14].

Women between 15 and 45 years of age with Diagnostic Related Groups codes 370–375 (caesarean and vaginal deliveries) were included. Deliveries occurring before 22 week’s gestation were excluded. Among all discharge reports, women with multiple pregnancies were identified using International Classification of Diseases, Ninth Revision, Clinical Modification (ICD-9-CM) codes (651.0x, 651.1x, 651.2x, 651.3x, 651.4x, 651.5x, 651.6x, 651.7x, 651.8x, 651.9x, V27.2, V27.3, V27.4, V27.5, V27.6 and V27.7). In particular, twin pregnancies were identified using 651.0x, 651.3x and V27.2, V27.3 and V27.4 ICD-9-CM codes.

Cases of GDM were established using ICD-9-CM code 648.8x. Throughout the study period, universal screening for GDM in Spain was based on a two-step strategy (O’Sullivan test and 100 g OGTT) following the recommendations of the Spanish College of Obstetricians and Gynecologists, Spanish Diabetes Society and Spanish Paediatrics Association [15]. The pre-existing diabetes group included women with type 1 diabetes mellitus (ICD-9-CM codes 250.x1 or 250.x3) and those with “type 2 diabetes mellitus and other pre-existing diabetes” (counting women with type 2 diabetes ICD-9-CM codes 250.x0 or 250.x2 and women with “diabetes mellitus, pre-pregnancy” codes 648.0x).

Maternal characteristics included age, chronic hypertension (ICD-9-CM codes 642.0–642.2 or 401–405), dyslipidaemia (ICD-9-CM code 272) and smoking status (ICD-9-CM code 649.0 or 305.1). In relation to obstetric complications, pre-eclampsia was defined by ICD-9 codes 642.4–642.6 and caesarean section by ICD-9-CM codes 74.0, 74.4, 74.9, 74.91, 74.99, 669.7 and 669.71 listed anywhere on the discharge report.

Perinatal outcomes analysed included stillbirth, preterm birth, mean birth weight, macrosomia, large-for-gestational-age (LGA) and small-for-gestational-age (SGA). Stillbirth or foetal mortality was considered if one or both twins (>22 week’s gestation) died in utero and was defined by ICD-9-CM codes V27.3 and V27.4 in twins and by V27.1 code in singleton pregnancies. Preterm delivery was defined as birth before 37 weeks’ gestation according to the World Health Organization (WHO) and the American College of Obstetricians and Gynecologists (ACOG). Macrosomia was defined according to the American College of Obstetricians and Gynecologists as newborns with a birthweight ≥4000 g. LGA and SGA were defined as neonatal birth weight >90th and <10th centiles for gestational age, respectively, based on singleton or twin birth weights standardised for foetal sex and gestational age using Catalonian population standards [16]. The prevalence of outcomes that included birth weight was calculated from the total of newborns.

Data from all delivery reports remained anonymous and therefore informed consent was not recorded. The study was conducted according to the principles of the Declaration of Helsinki and approved by the Clinical Research Ethics Committee of our institution (CEIC-Parc de Salut MAR, number 2017/7209/I).

### Statistical Analysis

Maternal and perinatal characteristics were reported using descriptive analysis. Chi-square test was used to compare prevalences of GDM and pre-existing diabetes among singleton and twin pregnancies. ANOVA was used to compare continuous variables among the three different glycaemic status groups. Maternal characteristics and perinatal outcomes of different study groups were compared using multivariate logistic regression models. Perinatal outcomes were adjusted for maternal age, hypertension, dyslipidaemia, year of delivery and smoking status. To study the interaction between singleton/twin pregnancies and glycaemic status on adverse outcomes the following analysis was undertaken. Firstly, adjusted Odds Ratio (OR) for each outcome in singleton vs. twin pregnancies regardless of glycaemic status, adjusted OR for GDM vs. non-DM and adjusted OR for pre-existing DM vs. non-DM in twin and singleton pregnancies were calculated. Secondly, a cross-product term between twin pregnancies and glycaemic status was included in multivariate models to assess the interaction effect of both factors and the risk of perinatal outcomes. In order to analyse the effect of each factor and its interaction on a specific adverse outcome, a B coefficient (and his statistical significance) was reported. Crude and age-adjusted annual GDM and pre-existing diabetes prevalences among twin pregnancies were calculated using direct standardisation to the maternal age structure of the whole study population. Time trends in twin pregnancies and the prevalence of GDM and pre-existing diabetes were assessed using a Poisson regression model. All *p*-values were two-tailed and statistical significance was accepted at 5% level. Data were analysed with the statistical software package IBM SPSS Statistics V.25.0.

## 3. Results

A total of 760,209 women gave birth in Catalonia between January 2006 and December 2015. Of these, 743,762 (97.8%) had singleton pregnancies. Data on singleton pregnancies, regarding prevalence and trends in perinatal outcomes of women with GDM, pre-existing diabetes and without diabetes, have been published elsewhere [3,4].

The vast majority of multiple deliveries (16,447 (2.16%)) corresponded to twin pregnancies (15,956 (2.10%)). We therefore analysed data on twin pregnancies, excluding other multiple deliveries. The rate of twin deliveries rose from 1.75% in 2006 to 2.22% in 2015 (*p* < 0.001).

Women with twin pregnancies were older than women with singleton gestations (33.7 ± 5.1 years vs. 31.4 ± 5.4 years, *p* < 0.001). Mothers of twins showed higher rates of chronic hypertension (0.9% vs. 0.6%, *p* < 0.001) and comparable rates of dyslipidaemia (0.2% in twins vs. 0.1% in singleton, *p* = 0.059). However, women with twin pregnancies were less frequently smokers than women with singleton pregnancies (4.0% vs. 5.6%, *p* < 0.001).

Among women with twin pregnancies, 14,785 (92.67%) were normoglycaemic, 1088 (6.82%) were diagnosed with GDM and 83 (0.52%) had pre-existing diabetes. Women with pre-existing diabetes included 63 (0.39%) with “type 2 diabetes mellitus and other pre-existing diabetes” and 20 (0.13%) with type 1 diabetes mellitus. The prevalence of GDM was higher in twin vs. singleton pregnancies whereas no differences were observed in pre-existing diabetes (Figure 1).

The prevalence of GDM increased among twin pregnancies during the study period (from 6.01% to 8.48%, *p* < 0.001). The prevalence of pre-existing diabetes in twin pregnancies remained stable (*p* = 0.416) (Figure 2).

### 3.1. Pregnancy Outcomes of Twin Pregnancies According to Glycaemic Status

Maternal characteristics and perinatal outcomes of twin pregnancies by maternal diabetes status are shown in Table 1.

In twin pregnancies, diabetes (either pre-existing or GDM) was associated with older mean maternal age and higher rates of pre-existing hypertension compared to non-diabetic pregnancies. The risk of pre-eclampsia was greater in twin pregnancies with pre-existing diabetes (15.66%) and GDM (11.39%) compared to normoglycaemic twin pregnancies (7.55%). By contrast, diabetes did not increase the risk of caesarean section; in particular, the caesarean section rate of GDM women was lower than non-diabetic women (65.63% vs. 68.12%, *p* = 0.002). The risk of preterm birth was higher in women with pre-existing diabetes compared with normoglycaemic women, although no statistically-significant differences were observed for women with GDM. The risk of stillbirth was not increased in women with diabetes (both pre-gestational and GDM) compared to normoglycaemic twin gestations.

Regarding birth weight outcomes in twin pregnancies, newborns of mothers with pre-existing diabetes were at increased risk of LGA (OR: 1.98) compared to those of non-diabetic mothers, while no statistically significant differences were observed in those of women with GDM. Moreover, no differences were found in rates of SGA, mean birthweight and macrosomia in neonates of women with diabetes (either pre-existing or GDM) compared to those of normoglycemic mothers.

### 3.2. Interaction between Twin Gestations and Glycaemic Status in Relation to Adverse Perinatal Outcomes

The effect of twin gestations, glycaemic status and the interaction of both conditions on adverse perinatal outcomes are shown in Table 2. Data on prevalence of diabetes and rates of perinatal outcomes in women according to their glycaemic status in singleton pregnancies have been published elsewhere [3,4].

The risk of pre-eclampsia was strongly increased in twin gestations (OR 5.38) and to a lesser extent in singleton pregnancies with pre-existing DM (OR 3.95) and GDM (OR 1.70). An attenuating effect of the interaction between twin pregnancies and pre-existing diabetes was observed (B coefficient −0.63, *p* = 0.045); however, women with pre-existing diabetes and twin pregnancies continued to have an increased risk of pre-eclampsia compared to non-DM. No significant interaction was detected between twin pregnancies and GDM.

In reference to preterm birth, the risk was increased in twin pregnancies compared to singleton (OR:16.8) and in singleton gestations with pre-existing diabetes (OR 3.17) or GDM (OR 1.2). A negative interaction between twin gestations and GDM was detected; as a result, their risk of preterm birth was not increased compared to non-DM twin pregnancies.

The risk of caesarean section was increased in twin pregnancies (OR 5.4) and in singleton gestations with pre-existing diabetes (OR 3.17) and GDM (OR 1.2). Moreover, an attenuating effect in caesarean section risk was observed between twin gestations and pre-existing DM and GDM. This attenuating effect translated into similar risks for pre-existing diabetes and normoglycemic twin pregnancies whereas GDM women had a lower risk.

The risk of LGA was reduced in twin pregnancies (OR 0.87) and increased in singleton pregnancies with pre-existing diabetes (OR 3.69) and GDM (OR 1.53). The attenuating effect observed between twin pregnancies and pre-existing DM and GDM resulted in similar rates of LGA in GDM and normoglycemic women although women with pre-existing diabetes and twin pregnancies continued to have an increased risk of LGA compared to non-DM.

The risk of SGA was decreased in twin pregnancies (OR 0.93) and to a greater extent in singleton pregnancies with GDM (OR 0.87) and pre-existing DM (OR 0.58). We detected an enhancing effect between twin gestations and GDM; therefore, no differences were observed in SGA rates between twins with GDM and without DM.

No interaction was detected between twin pregnancies and glycaemic status for macrosomia and stillbirth.

## 4. Discussion

This large, population-based study revealed a 27% rise in twin pregnancies and a 41% increase in GDM prevalence among twin gestations in Catalonia from 2006 to 2015. The prevalence of GDM in twin pregnancies was 6.82%, clearly higher than the prevalence of GDM in singleton gestations (4.80%). Regarding obstetric and perinatal outcomes in twin pregnancies, diabetes, both pre-existing and GDM, were associated with an increased risk of pre-eclampsia compared to normoglycaemic women. Prematurity and LGA were more frequent in women with pre-existing diabetes than in non-diabetic mothers. However, the impact of both, pre-existing diabetes and GDM, on twin pregnancy outcomes was attenuated when compared with its impact on singleton gestations. To our knowledge, no previous study has analysed the interaction of both, twin pregnancy and diabetic conditions, on pregnancy outcomes.

The twin pregnancy rate observed in our study was consistent with the 1.4–3% rates reported in previous studies [12,17,18,19,20]. The increasing rate of twin pregnancies observed has also been described worldwide over recent decades [12]. In particular, a multi-centre-based study conducted in Spain reported an increase in the rate of twin gestations from 1.69% in 2000 to 2.06% in 2004 [17]. The principal factors related to this trend are delayed childbearing and the rise in ART. Advanced maternal age is a known risk factor for spontaneous twin pregnancies [12]. In this respect, it has been proposed that, under equal ovarian feedback, premenopausal mothers of dizygotic twins have hyper-stimulation by endogenous follicle-stimulating hormones [21]. The use of ART treatments steadily increased in Spain over the study period. A change in embryo transference policies in the recent decades, with a change from multiple to single embryo transferences, has led to a lower rate of multiple pregnancies among women receiving ART treatments [22]. However, the overall number of ART treatments and twin pregnancies has continued to rise which, together with older mean maternal age, might account for the trend in twin pregnancy rates observed in our study.

Previous studies described GDM prevalences in twin pregnancies ranging from 3.0% to 9.0% [13,18,19,23,24,25,26]. Lai et al. conducted a population-based study in Alberta, Canada between 2003–2014 and found a 7.3% prevalence of GDM among twin pregnancies, slightly higher than the 6.8% prevalence observed in our study [18]. Furthermore, Gonzalez et al. reported a 7.7% prevalence in a multi-centre-based study in Spain where medical reports between 2004 and 2008 were reviewed. Data from two hospitals in Catalonia were included and the same GDM diagnostic approach as in our study was used [27].

The prevalence of GDM was higher in twin than in singleton pregnancies in the present study. This finding is in line with most previous studies [13,18,28,29,30] although some authors reported conflicting results on the subject [31,32]. Although factors associated with twin pregnancies such as maternal age, obesity, increased weight gain and the use of ART might have played a role in the higher prevalence observed, a study by Rauh-Hain et al. found twin pregnancy to be associated with a two-fold increase in the risk of GDM, after adjusting for maternal age, ethnicity, body mass index, high blood pressure, smoking and parity, suggesting that twin pregnancy itself might be an independent risk factor for GDM [33]. In this respect, it has been suggested that larger placental mass and higher levels of human placenta lactogen, oestrogen and progesterone in twin pregnancies might increase the risk of GDM [34,35].

A growing trend of GDM among twin pregnancies has also been described in previous studies. In a population-based study conducted in Finland which analysed > 23,000 twin pregnancies, the prevalence of GDM in twin pregnancies rose from 3.3% in 1987 to 20.7% in 2014 [20] and a three-fold higher prevalence was observed in a single-centre Australian study between 2003 and 2012 [24]. This increasing trend is in line with the rise in GDM prevalence observed worldwide [6] and in particular with that observed in our population study in Catalonia [3].

Few studies assessed the prevalence of pre-existing diabetes in twin pregnancies. Lai et al. reported an 0.8% prevalence of pre-existing diabetes in twin pregnancies and an 0.9% prevalence in singleton gestations [18]. However, Gonzalez et al. found a lower prevalence of pre-existing diabetes in twins of 0.025% [36]. To our knowledge, the present study is the first one to report the trend in prevalence of pre-existing diabetes among twin pregnancies, and the results show no change overtime during the study period.

Regarding hypertensive complications in twin pregnancies, women with a diabetic condition were found to have higher rates of pre-eclampsia; 15% for those with pre-existing diabetes (odds ratio: 2.11) and 11% for those with GDM (odds ratio: 1.45). This finding is in line with several previous studies [18,24,27]; however, evidence on the effect of diabetes on hypertensive disorders in twin pregnancies is mixed. Dinham et al. found a 19.8% prevalence of hypertensive disorders in GDM compared to the 11.6% rate observed in normoglycaemic twin gestations [24]. Lai et al. also observed an increased risk of pre-eclampsia in women with GDM, but no significant differences in pre-existing diabetes [18]. Okby et al. reported an increase in mild pre-eclampsia in GDM (10.7% vs. 5.2%, *p* < 0.001), but not in severe pre-eclampsia (4.6% vs. 3.3%, *p* = 0.18) in their cohort of twin pregnancies [19]. On the other hand, Mourad et al. observed that the increased risk of hypertensive disorders in twin gestations with GDM lost significance after adjustment for maternal age, in vitro fertilisation treatment and pre-pregnancy BMI [37]. Furthermore, Hiersch et al. concluded that, unlike in singleton pregnancies, GDM in twins was not associated with hypertensive complications [13]. In the present study, we found an attenuation in the impact of pre-existing diabetes in the rate of pre-eclampsia in twin pregnancies, not observed in women with GDM.

In our study, diabetes did not raise the risk of caesarean section in twin pregnancies. While some previous studies reported an increased risk of caesarean section in women with diabetes and twin pregnancies [13,33,37], others found no significant differences [23,26,27,38]. In a retrospective population-based study conducted from 1998 to 2010 in Israel, Okby et al. observed a higher risk of caesarean section for GDM twin pregnancies; however, GDM was not found to be an independent risk factor for caesarean section in a multivariable analysis (controlling for maternal age, fertility treatments and hypertensive disorders) [19]. The similar rate of caesarean section observed in twin pregnancies with diabetes contrasts with the findings in singleton pregnancies. In this respect, twin pregnancies have a high base line risk of caesarean section and thus the effect of diabetes might be less relevant. Moreover, the increased risk of caesarean section in singleton pregnancies with diabetic conditions might be partly driven by high rates of macrosomia. In this respect, as described in the present study, macrosomia very rarely complicates twin pregnancies. It has been suggested that this might explain the similar rates of caesarean section in GDM and normoglycaemic twin gestations [19].

In the present study, the risk of preterm birth in twin pregnancies was increased in women with pre-existing diabetes. Previous studies have yielded consistent results [18,36,39]. However, in contrast with the results in singleton pregnancies, no differences in prematurity rates were observed in twin pregnancies with GDM. Previous studies evaluating GDM reported mixed results. Gonzalez et al. found an increased risk of preterm birth in women with GDM and twin pregnancies, although after adjusting for potential confounders, the presence of diabetes did not appear to significantly influence prematurity [27]. Hiersch et al. and Luo et al. reported higher prematurity rates in women with GDM and twin pregnancies, but the increased risk was less marked compared to the results observed in singleton [13,30]. In line with our findings, other previous studies did not report an increased risk of prematurity in women with twin pregnancies with GDM [18,19,24]. Women with twin pregnancies, regardless of glycaemic status, had a very high rate of prematurity (almost 50%). The fact that GDM increases the risk of prematurity in singleton pregnancies might be related, at least in part, with excessive foetal growth, a situation that does not occur in GDM twin pregnancies.

In the present study cohort, diabetes was not associated with an increased risk of stillbirth in twin pregnancies, a finding which is consistent with prior evidence [18,19,24,27,33,39]. However, it contrasts with the results observed in singleton pregnancies, with increased risk of stillbirth in women with pre-existing diabetes and a reduction in the risk in women with GDM. While the increased risk of stillbirth in women with pre-existing diabetes is widely accepted, prior evidence related to stillbirth and GDM in singleton pregnancies is conflicting. Nevertheless, other population-based studies similarly described a reduction in risk of stillbirth in these women [11,18].

Our results show that twin gestations with pre-existing diabetes were at increased risk of LGA compared to non-diabetic twin gestations. This finding is consistent with previous studies [18,30,39]. The increasing effect of pre-existing diabetes on LGA was attenuated in our cohort of twins. In this respect, Luo et al. consistently reported a higher risk of LGA in women with diabetes mellitus and singleton pregnancies (OR 1.89) than in twin gestations (OR 1.21) [30].

In regard to the effect of GDM on birthweight outcomes, previous evidence is conflicting. In our study, we observed no effect of GDM in the risk of LGA and SGA. These findings are in line with prior evidence on the risk of LGA [24,27,38] and SGA [13,18,24,26] and contrast with the results observed in singleton pregnancies. In this context, it has been suggested that gestational diabetes might not have a large impact on foetal overgrowth [38]. These findings, together with the fact that macrosomia is very rare in twin pregnancies, has led some authors to propose potential benefits of mildly-elevated glucose levels on foetal growth, compensating for other growth restricting circumstances observed in twin pregnancies [13,30,38]. In fact, a meta-analysis conducted by McGrath et al. found no association between GDM in twin pregnancies and serious adverse perinatal outcomes [25]. Moreover, Luo et al. reported a reduced risk of low 5-min Apgar score and neonatal death in women with DM and twin pregnancies [30]. The reduction in risk of low 5-min Apgar score and perinatal mortality was also observed by Okby et al. [19]. Nevertheless, other authors alert to the potential harm of hyperglycaemia regarding other maternal and perinatal outcomes as well as potential long-term metabolic complications [13,27]. In this respect, it might be hypothesized that these improved results are related to a more intensive follow-up and care in women with GDM and twin gestations. The higher maternal age in the GDM group could be associated with higher ART treatment rates, which in turn, might contribute to the compliance with pregnancy follow-up.

Evidence on the impact of GDM treatment in twin pregnancies is scant and shows mixed results. Fox et al. reported that improved glycaemic control in twin pregnancies with GDM was not related to improved outcomes and was associated with a higher risk of SGA [23]. Unfortunately, recommendations on the diagnosis and treatment of gestational diabetes are based on randomised controlled trials that excluded twin pregnancies or only included a small number of them [9,40,41]. Therefore, prospective studies on twin gestations are needed to assess a specific diagnostic approach, glycaemic control targets and management guidelines in twin pregnancies with DM.

The main strengths of our study lie in the population-based design, using a validated and widely-used database, and the large sample size. We analysed data from >750,000 deliveries and >15,000 twin deliveries over a ten-year period. It should be outlined that, unlike other previous studies, we were able to distinguish between gestational diabetes and pre-existing diabetes and we innovatively analysed the interaction between twin gestation and glycaemic status. Moreover, no changes were made in the diagnostic GDM protocol over the study period, allowing us to assess trends in the prevalence of GDM in twin pregnancies. However, this study did have several limitations. First, we conducted a retrospective analysis of an administrative database and diagnoses were based on ICD-9-CM codes with potential bias related to validity and accuracy of coding. Unfortunately, this study lacked data on maternal BMI, ethnicity, ART rates, level of glycaemic control and chorionicity. Furthermore, we were not able to analyse major perinatal outcomes such as neonatal and perinatal mortality, congenital anomalies and newborn intensive care unit admission since linked maternal and neonatal data were not available. Moreover, we were unable to control for unmeasured confounders such as increased maternal BMI, ethnicity and other socioeconomic factors Finally, although universal screening for GDM is recommended in Spain, published data on actual screening rates are lacking.

In conclusion, prevalence of GDM in twin pregnancies increased between 2006 and 2015 in Catalonia whereas the prevalence of pre-existing diabetes remained stable. In our study cohort, both GDM and pre-gestational diabetes entailed an additional risk of pre-eclampsia and the latter also increased the risk of prematurity and LGA. However, the effect of diabetes on adverse perinatal outcomes was attenuated in twin gestations when compared with singleton pregnancies. In the light of these results, it should be questioned if women with twin pregnancies should receive the same treatment for their diabetic condition as those with singleton pregnancies. Population-based data provide helpful information on risk of perinatal outcomes; however, randomised studies evaluating the treatment of these conditions in twin pregnancies are required to help guide the management of diabetes in these women.

## Figures and Tables

**Figure 1 jcm-10-01937-f001:**
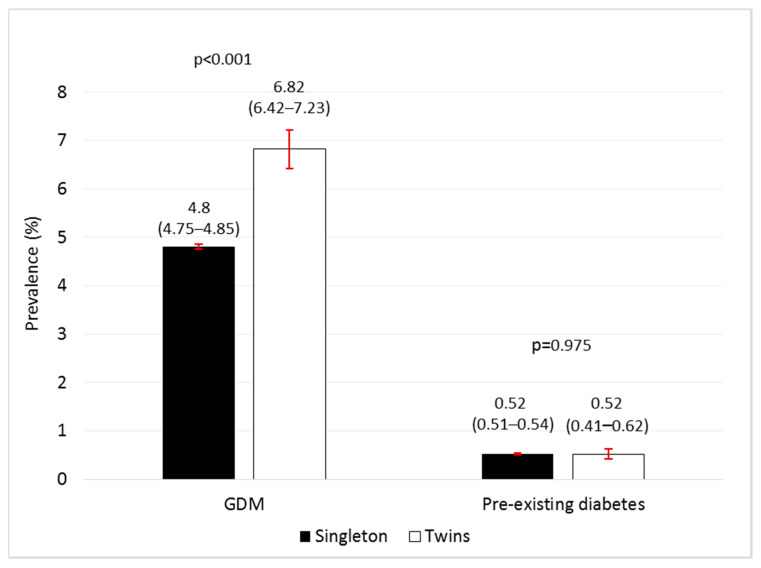
Prevalence (95% CI) of gestational diabetes mellitus (GDM) and pre-existing diabetes in twin and singleton pregnancies in Catalonia between 2006 and 2015.

**Figure 2 jcm-10-01937-f002:**
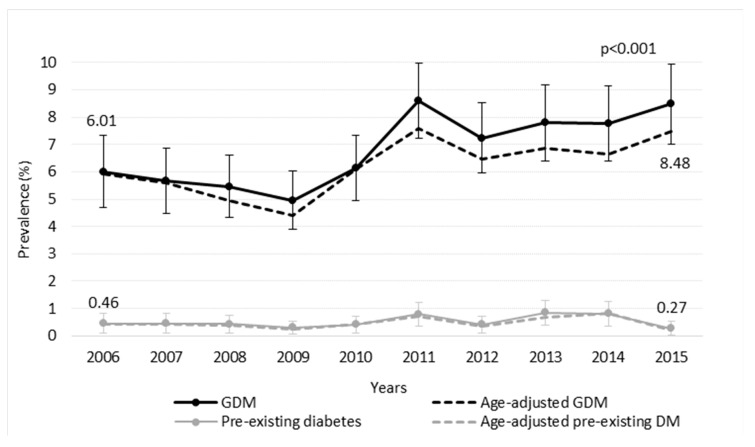
Trends in crude and age-adjusted prevalence (95% CI) of GDM and other pre-existing diabetes in twin pregnancies in Catalonia between 2006 and 2015.

**Table 1 jcm-10-01937-t001:** Maternal characteristics, obstetric and perinatal outcomes of twin pregnancies.

	Normoglycaemic *n*= 14,785	GDM *n* = 1088	OR, GDM vs. Non-DM	Pre-Existing DM *n* = 83	OR, Pre-Existing DM vs. Non-DM	ANOVA *p* Value
Age (years)	33.6 ± 5.1	35.3± 5.0		35.1 ± 6.1		*p* < 0.001
Chronic hypertension, *n* (%)	127 (0.86)	18 (1.65)	1.81 (1.10–2.99) *p* = 0.020	3 (3.61)	4.09 (1.27–13.14) *p* = 0.018	
Dyslipidaemia, *n* (%)	26 (0.18)	2 (0.18)	1.05 (0.25–4.44) *p* = 0.949	1 (1.20)	7.06 (0.94–52.78) *p* = 0.057	
Smoking, *n* (%)	583 (3.94)	54 (4.96)	1.40 (1.05–1.87) *p* = 0.002	3 (3.61)	0.98 (0.31–3.11) *p* = 0.969	
Pre-eclampsia, *n* (%)	1116 (7.55)	124 (11.39)	1.43 (1.17–1.74) *p* < 0.001	13 (15.66)	2.04 (1.12–3.72) *p* = 0.020	
Preterm birth, *n* (%)	7373 (51.84)	566 (53.04)	1.06 (0.94–1.21) *p* = 0.327	55 (69.62)	2.15 (1.33–3.48) *p* = 0.002	
Caesarean section, *n* (%)	10,072 (68.12)	714 (65.63)	0.82 (0.71–0.93) *p* = 0.002	61 (73.49)	1.20 (0.73–1.97) *p* = 0.468	
Stillbirth, *n* (%)	473 (3.20)	26 (2.39)	0.76 (0.51–1.14) *p* = 0.189	2 (2.41)	0.77 (0.18–3.31) *p* = 0.712	
Macrosomia, *n* (%)	22 (0.08)	1 (0.05)	0.595 (0.08–4.43) *p* = 0.612	0 (0)	0 *p* = 0.966	
LGA (>90th), *n* (%)	3247 (11.63)	273 (12.95)	1.12 (0.98–1.28) *p* = 0.09	33 (20.89)	1.98 (1.35–2.91) *p* = 0.001	
SGA (<10th), *n* (%)	2339 (8.37)	202 (9.58)	1.15 (0.99–1.34) *p* = 0.078	8 (5.06)	0.58 (0.28–1.19) *p* = 0.134	
Mean birth weight (g)	2333 ± 503	2336 ± 486		2295 ± 616		*p* = 0.304

GDM, gestational diabetes; DM, diabetes mellitus; Non-DM, non-diabetic; LGA, large for gestational age; SGA, small for gestational age. ANOVA was used to compare age and mean birthweight among the three groups. Hypertension and dyslipidaemia OR were adjusted for maternal age and smoking habit. Pre-eclampsia, preterm birth, caesarean section, stillbirth, macrosomia, LGA and SGA OR were adjusted for maternal age, chronic hypertension, dyslipidaemia, year of delivery and smoking status. Birthweight outcomes were calculated using the total number of living newborns with weight data (27,931 normoglycaemic, 2108 GDM and 158 pre-existing DM). Preterm birth was calculated considering the cases with gestational age at delivery data (14,222 normoglycaemic, 1067 GDM, 79 pre-existing DM). The other outcomes were calculated using the total number of deliveries in each group.

**Table 2 jcm-10-01937-t002:** Interaction between twin pregnancies and glycaemic status on adverse perinatal outcomes.

	Twins vs. Singleton		Gestational Diabetes				Pre-Existing Diabetes			
	B Coefficient Normoglycaemic Pregnancies *p* Value	OR Twins vs. Singleton *p* Value	B Coefficient Singleton Pregnancies *p* Value	OR GDM vs. Non-DM (Singleton) *p* Value	OR GDM vs. Non-DM (Twins) *p* Value	Interaction GDM and Twins B Coefficient *p* Value	Beta Coefficient Singleton Pregnancies *p* Value	OR Pre-Existing DM vs. Non-DM (Singleton) *p* Value	OR Pre-Existing DM vs. Non-DM (Twins) *p* Value	Interaction Pre-Existing DM and Twins B Coefficient *p* Value
Pre-eclampsia	1.68 *p* < 0.001	5.38 (5.04–5.74) *p* < 0.001	0.532 *p* < 0.001	1.70 (1.59–1.82) *p* < 0.001	1.43 (1.17–1.74) *p* < 0.001	−0.12 *p* = 0.275	1.37 *p* < 0.001	3.95 (3.44–4.51) *p* < 0.001	2.04 (1.12–3.72) *p* = 0.020	−0.63 *p* = 0.045
Preterm birth	2.82 *p* < 0.001	16.8 (16.26–17.43) *p* < 0.001	0.18 *p* < 0.001	1.20 (1.15–1.25) *p* < 0.001	1.06 (0.94–1.21) *p* = 0.327	−0.15 *p* = 0.028	1.15 *p* < 0.001	3.17 (2.91-–3.44) *p* < 0.001	2.15 (1.33–3.48) *p* = 0.002	−0.429 *p* = 0.085
Caesarean section	1.73 *p* < 0.001	5.64 (5.44–5.84) *p* < 0.001	0.086 *p* < 0.001	1.09 (1.06–1.12) *p* < 0.001	0.82 (0.71–0.93) *p* = 0.002	−0.30 *p* < 0.001	0.791 *p* < 0.001	2.20 (2.07–2.35) *p* < 0.001	1.20 (0.73–1.97) *p* = 0.468	−0.621 *p* = 0.015
Stillbirth	1.92 *p* < 0.001	6.85 (6.21–7.57) *p* < 0.001	−0.305 *p* = 0.001	0.74 (0.62–0.88) *p* < 0.001	0.76 (0.51–1.14) *p* = 0.189	−0.025 *p* = 0.912	0.73 *p* < 0.001	2.08 (1.52–2.83) *p* < 0.001	0.77 (0.18–3.31) *p* = 0.712	−1.072 *p* = 0.145
Macrosomia	−4.406 *p* = 0.012	0.012 (0.008–0.019) *p* < 0.001	0.414 *p* < 0.001	1.51 (1.45–1.57) *p* < 0.001	0.595 (0.08–4.43) *p* = 0.612	−0.905 *p* = 0.376	0.99 *p* < 0.001	2.68 (2.45–2.94) *p* < 0.001	0 *p* = 0.966	−15.037 *p* = 0.996
LGA	−0.14 *p* < 0.001	0.87 (0.84–0.90) *p* < 0.001	0.43 *p* < 0.001	1.53 (1.49–1.58) *p* < 0.001	1.12 (0.98–1.28) *p* = 0.09	−0.302 *p* < 0.001	1.31 *p* < 0.001	3.69 (3.46–3.95) *p* < 0.001	1.98 (1.35–2.91) *p* = 0.001	−0.610 *p* = 0.002
SGA	−0.07 *p* = 0.001	0.93 (0.89–0.97) *p* < 0.001	−0.137 *p* < 0.001	0.87 (0.83–0.91) *p* < 0.001	1.15 (0.99–1.34) *p* = 0.078	0.27 *p* = 0.001	−0.55 *p* < 0.001	0.58 (0.50–0.60) *p* < 0.001	0.58 (0.28–1.19) *p* = 0.134	−0.01 *p* = 0.98

GDM, gestational diabetes; Non-DM, non-diabetic; LGA, large for gestational age; SGA, small for gestational age. OR were adjusted for maternal age, chronic hypertension, dyslipidaemia, year of delivery and smoking status.

## Data Availability

Data are available upon request from the authors.

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
