# Peer review of "Trends in Prevalence of Diabetes among Twin Pregnancies and Perinatal Outcomes in Catalonia between 2006 and 2015: The DIAGESTCAT Study"

_jcm, 2021, doi:10.3390/jcm10091937_

Round 1
Reviewer 1 Report
My queries have been addressed. thank you.
Author Response
Thank you
Reviewer 2 Report
The present study represents a population-based study aiming to assess the effects of diabetes mellitus in perinatal outcomes of twin gestations. The following issues should be taken into account in order to improve the quality of the manuscript.
-The exact study design should be reported (ie. retrospective cohort?).
-It is stated that " universal screening for GDM in Spain was based on a two-step strategy". However, this is just the recommendation and it is assumed that it had applied for all participants. The possibility of misclassification bias cannot be safely excluded.
-It should be specified whether "hypertension" referrs to chronic hypertension or gestational hypertension.
-In order to limit the inherent confounding of the study due to its observational character, it is suggested to perform propensity score matching aiming to evaluate the effects of diabetes in perinatal outcomes among twin pregnancies.
-Error bars should be added in Figure 1 and 2.
-A pathogenetic rationale of the differential effects of diabetes mellitus in signleton and twin pregnancies may be collaborated in the Discussion section.
Author Response
1.The exact study design should be reported (ie. retrospective cohort?).
Response: In accordance with the reviewer’s suggestion, we specified that the study conducted was a retrospective cohort study (see Methods, line 75).
2. It is stated that " universal screening for GDM in Spain was based on a two-step strategy". However, this is just the recommendation and it is assumed that it had applied for all participants. The possibility of misclassification bias cannot be safely excluded.
Response: We agree with the reviewer’s comment, we added this potential bias as a limitation of the study (see Discussion, line 443).
3. It should be specified whether "hypertension" refers to chronic hypertension or gestational hypertension.
Response: The variable “hypertension” was defined by ICD-9-CM codes that refer to pre-existing hypertension (642.0‐642.2 or 401‐405). In agreement with the reviewer’s recommendation, we specified that the variable hypertension refers to chronic hypertension.
4. In order to limit the inherent confounding of the study due to its observational character, it is suggested to perform propensity score matching aiming to evaluate the effects of diabetes in perinatal outcomes among twin pregnancies.
We appreciate the reviewer’s comment. However, although we have not conducted propensity score matching, to evaluate the effects of diabetes in perinatal outcomes among twin pregnancies, we performed a multivariate analysis, adjusting for confounding variables (maternal age, hypertension, dyslipidaemia, year of delivery and smoking status) following the first reviewer’s recommendation.
5. Error bars should be added in Figure 1 and 2.
Response: We appreciate the reviewer’s comment and we have added error bars to Figure 1 and Figure 2.
6.A pathogenetic rationale of the differential effects of diabetes mellitus in signleton and twin pregnancies may be collaborated in the Discussion section.
Response: Given that causal relationships cannot be established in the light of this study’s findings, several hypothesis on the differential effects of diabetes in twin pregnancies have been made through the Discussion section. First of all, twin pregnancies have very high base-line risk rates of certain perinatal outcomes such as Caesarean section, preterm birth and pre-eclampsia, that might mask the effect of diabetes (see Discussion, line 384). Moreover, the effect of diabetes in the rates of Caesarean section is partly related to the high rates of macrosomia (not present in twin pregnancies) (Discussion, line 370). In this respect, factors leading to Caesarean section in twins are varied; however, not often related to excessive birthweight. Furthermore, as it has been proposed by other authors, in the Discussion of the article (line 408), we suggested that the effects of foetal growth restriction in twins might be counterbalanced by the excessive growth observed in babies of diabetic mothers. Finally, and more importantly, it is possible that women with diabetes and twin pregnancies might receive a more intensive follow up, more frequent medical interventions and have a higher compliance to visits, therefore affecting the observed the perinatal outcomes (Discussion, line 417). After revising previous literature, we have not been able to come up with other hypothesis that might explain the different effects of diabetes in twin pregnancies. In accordance with the reviewer’s suggestion, we have emphasized the potential role of follow up and care in twin pregnancies with diabetes (Discussion, line 418).
Round 2
Reviewer 2 Report
The authors have adequately revised their manuscript following the reviewers' comments. Before it can be accepted for publication, the following points should be addressed:
-English language should be improved by correcting all typos (e.g. line 412 "alternativa")
-The possibility of unmeasured confounding should be acknowledged as a limitation.
-In the figures legends, it should be specified what the error bars indicate (e.g. SD, SE).
Author Response
Comment 1: Some typographical errors have been corrected. The paragraph containing "alternativa" has been removed.
Comment 2: Limitations section has been updated following reviewer suggestion.
Comment 3: Error bars indicate 95% CI.
This manuscript is a resubmission of an earlier submission. The following is a list of the peer review reports and author responses from that submission.
Round 1
Reviewer 1 Report
I find that the article is difficult to interpret and to read. I think that is because there are two different parts of the study
1- First part: considering both twin and singleton pregnancies
2- Second part: considering only twin pregnancies
1- First part comparing twin and singleton pregnancies
1a percentage (and not prevalence) of twin pregnancies among all pregnancies (fig 1)
1b comparison of prevalence of GDM and preexisting diabetes (“diabetes”) in case of twin vssingleton pregnancies: more GDM; but similar prevalence of diabetes
how to explain that? The authors should try to explore this point even if there are not many variables in the CMBD-AH. Especially, BMI and ART method (if any) would be of interest (Carbillon et al. Eur J Obst Gynecol Reprod Biol 2017 137) but are lacking. Furthermore, we do not have any data about accuracy of coding for hypertension, dyslipidemia, smoking?
However, the % of the four available determinants (age smoking dyslipidemia hypertension) should be compared in twin vssingleton pregnancies. And the prevalence of GDM and diabetes could be adjusted for them.
2- Second part considering only twin pregnancies
2a- trends in prevalence of hyperglycemic states (both GDM and diabetes) by years: GDM increases; diabetes =
2b- the rate of adverse pregnancy outcomes:
GDM more preeclampsia, less caesarean section, similar prematurity LGA SGA BW
Diabetes more preeclampsia prematurity LGA, similar Caesarean section SGA BW
These results are of interest but would be far more interesting if the entire cohort would be considered. For example, in clinical practice, I would like to know whether the effect of glycemic status (normal vs GDM vs diabetes) is similar during singleton and twin pregnancies. This means comparing the rate of each outcome according to glycemic status in singleton pregnancies, then in twin pregnancies, with overall p value and if significant: p value for glycemic status, p value for S/Twin pregnancies and p value for interaction.
Furthermore, the results should be adjusted for available variables: age, smoking, hypertension, dyslipidemia (if you think that accuracy is correct); but also year of delivery as care has probably changed over time.
Some specific points:
Please use the same terminology throughout the manuscript for hyperglycemic states: preexisting diabetes and GDM for example. You use line 39 (L39) the term “diabetes in pregnancy” which corresponds in some guidelines to overt diabetes diagnosed during pregnancy and this is confusing. L16 “hyperglycemic states” rather than diabetes?…
Introduction
I think your points are not clearly stated. My point would be (but you may for sure disagree):
- Some studies have shown that twin pregnancies are of poor prognosis as compared to singleton pregnancies.
- Hyperglycemic states are also of poor prognosis (I propose you to add a recent study considering also health administrative data/ Billionnet et al. Diabetologia 2017 636).
- However, an interaction between both “conditions” could be present when considering adverse pregnancy outcomes. For example, after adjustment for BMI and age and considering only women with twin pregnancy, women with GDM have a similar rate of LGA infant as those without (Poulain Diabetes Metab 2015/ 387). This is clearly not the case during singleton pregnancies;
--> the question for my clinical practice would be “is the poor prognosis associated with hyperglycemic states during pregnancy similar in case of twin pregnancies as in singleton pregnancies?” (Is there an interaction?). In practice, should I speculate that I should care for women with hyperglycaemic states during twin pregnancies the same way as women during singleton pregnancies.
Methods
- Again, what about accuracy for smoking, hypertension, dyslipidemia codes?
- Not clear to me: prevalence of LGA and SGA. Are 100% of the newborns linked to mother? The number of newborns should be indicated in the Table before LGA SGA BW.
- Table 1 (which would anyway change if you consider my issues): 3 groups --> please consider global anova, then Bonferroni multiple comparisons
- Again for outcomes, consider also adjusting for year of delivery
Results
Fig 1: not really crucial / aims of the study --> supplementary data? This is the % / rate of twin pregnancies among all pregnancies and not the prevalence
L140_158: not clear whether crude or adjusted results
Discussion
Would be adapted with new results…
I am conscious that what I recommend means for you a very large amount of work but I do think these results would be far more relevant.
Reviewer 2 Report
This article describes trends in prevalence of diabetes in twin pregnancies and associated maternal outcomes in Catalonia between 2006-15. Population-based data are used.
This is a nice piece of work and important to publish as many diabetes in pregnancy studies exclude twin pregnancies.
The abstract should contain the absolute numbers of gestational and pre-gestational twin pregnancies
It seems a little odd that data are available for birthweight, but not for the birth outcome - such as live- or stillbirth. Can the authors comment a bit more on this fact.
Are the odds ratios in table 1 adjusted for maternal age and/or other variables. This is what seems to be described in the methods and if so, this should be clearly outlined in the table legend.
I recommend rechecking all the numbers for accuracy. One example of concern is table 1, LGA in preexisting diabetes, then N and % don't make sense.
In addition, I'm not convinced regarding the usefulness/appropriateness of some of the statistical tests. For example, there was 1 case of preexisting diabetes with dyslipidemia - hardly enough to do a statistical test.
There are a few minor issues such as introduction line 40, would be better to read "maternal and infant/neonatal outcomes" and line 45 would read better "in association with increasing maternal age"
